# Gravity-Compensation Design Approaches for Flexure-Pivot Time Bases

Etienne Thalmann , Quentin Gubler and Simon Henein *

**Abstract:** While flexure time bases have gained significant traction in the watchmaking industry thanks to their high quality factor and monolithic design, maintaining a stable frequency in varying orientations of wrist watches with respect to gravity remains a significant challenge. This results from the fact that the flexures play two roles simultaneously: guiding the oscillating mass along a one-degree-of-freedom pivotal motion, and providing the oscillator's elastic restoring force. Indeed, varying stress-stiffening effects induced by the varying direction of the weight of the oscillating mass affect the pivot angular stiffness, which impacts its oscillating frequency. In order to address this issue, two design approaches are presented which, when combined, allow to reach the strict chrono-metric standards of mechanical watches. Firstly, the frequency differences for all vertical positions (i.e., gravity orthogonal to the rotation axis) are mitigated by designing architectures with reduced parasitic center shift, or by offsetting the center of mass (COM) along their axis of symmetry, or both. Secondly, the frequency differences between vertical and horizontal positions (i.e., gravity parallel to the rotation axis) are reduced by offsetting the COM along the rotation axis. The implementation and effectiveness of these approaches are demonstrated by numeric simulations, as well as by experimental measurements performed on watch-scale silicon etched prototypes.

**Keywords:** compliant mechanism; flexure pivot; gravity compensation; oscillator; time base





## 1. Introduction

Flexure time bases have gained significant traction in the watchmaking industry in the past decade thanks to their high quality factor and monolithic design offering the prospect of increased timekeeping accuracy, increased power reserve, reduced maintenance and simplified assembly [1–4]. They have been the topic of our previous research [4–10], and recently gave rise to commercial products [11–13]. These mechanisms use the deformation of slender elastic elements, i.e., flexures, to guide their motion [14–16]. This offers the advantages that the motion is free of contact friction and opposed by a spring force, thus providing the conditions for high-quality oscillators. However, since the elements that provide the restoring force also provide guidance, one intrinsic limitation of flexure mechanisms is that their stiffness is affected by external loads, an effect known as stress stiffening [17–19]. This is a major issue for portable time bases such as mechanical watches, where changes in their orientation with respect to gravity affect loading conditions, and where oscillator stiffness variations directly deteriorate frequency stability, i.e., timekeeping accuracy.

In order to mitigate this issue, we provide a two-step process for compensating (and thus reducing) the impact of gravity on the stiffness of flexure-pivot oscillators:

1. Minimizing the difference in pivot stiffness for gravity orientations lying within the rotation plane (i.e., oscillator in vertical position with respect to gravity). Design approaches to mitigate these *in-plane* effects are described in Section 2.1.

2.　　Minimizing the difference between the mean pivot stiffness for gravity orientations within the rotation plane (step 1) and the stiffness when gravity acts along the rotation axis (i.e., oscillator in horizontal position with respect to gravity). Design approaches to mitigate these *out-of-plane* effects are described in Section 2.2.

It is then assumed that, after these two steps, the effect of gravity on stiffness will be minimized for all orientations of the flexure pivot with respect to gravity. This assumption is backed by the fact that these are second-order effects of the external load and that they can, hence, be decomposed into the sum of orthogonal components that are either in the rotation plane or perpendicular to it.

While some of the design approaches to reduce in-plane gravity effects have already been discussed partially in the literature [20] and in our previous research [4–6], the main contributions of this paper are:

- The novelty of the out-of-plane effects mitigation technique;
- The combination of both aforementioned steps to reach new levels of frequency stability;
- The evaluation of these approaches in terms of chronometric accuracy on designs and prototypes satisfying realistic mechanical watch specifications.

This is demonstrated on two flexure-pivot oscillator architectures developed in previous research: the co-RCC [4,6,8] (Figure 1a) and the TRIVOT [10,21] (Figure 1b). For the co-RCC oscillator, watch-scale silicon prototypes are used to validate experimentally the presented approaches. For the TRIVOT oscillator, a finite element model (FEM) of a watch-scale design is used, as no prototype has been manufactured yet. (See videos of the co-RCC in motion here: https://youtube.com/shorts/yGv_djjGGc0?feature=share accessed on 1 June 2022. See videos of the TRIVOT in motion here: https://youtu.be/rUH3cDmeXC4 accessed on 1 June 2022).

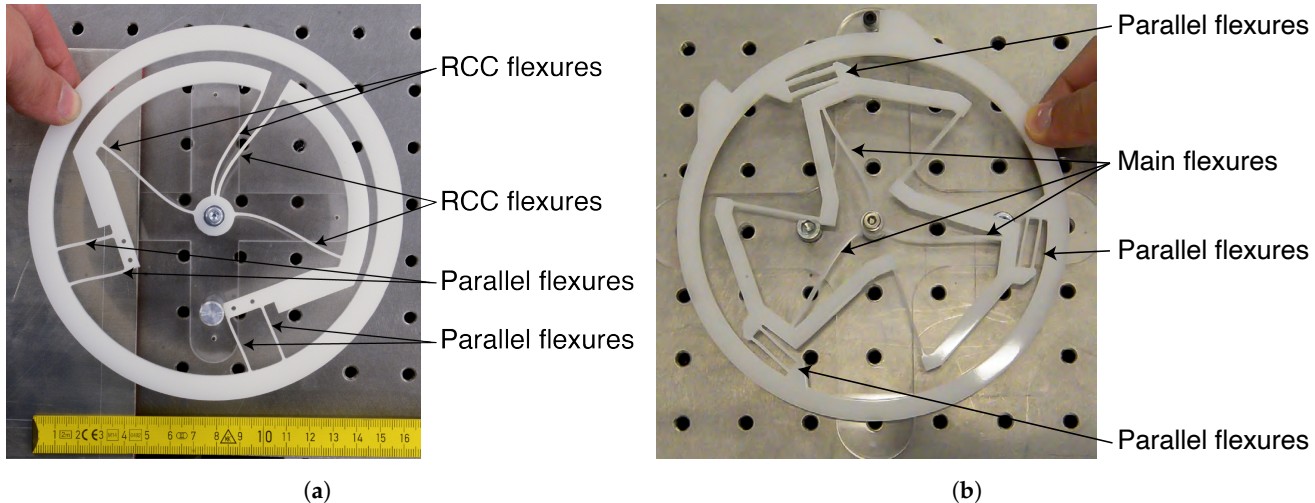

(**a**)　　　　　　　　　　　　　　　　　　　　(**b**)

**Figure 1.** Large-scale (150 mm diameter) mock ups of the (**a**) co-RCC and (**b**) TRIVOT flexure-pivot oscillators fabricated by laser cutting a 5 mm thick Polyoxymethylene (POM) sheet. They are shown in rotated position (respectively, 26 and 22 degrees from equilibrium position).

## 2. Gravity-Compensation Design Approaches

In this section, practical ways of reducing the influence of gravity on the rotational stiffness of the co-RCC and TRIVOT flexure pivots are detailed. They are implemented in two steps, by, firstly, mitigating the in-plane effects (Section 2.1) and, secondly, making them as close as possible to the out-of-plane effects (Section 2.2). All our solutions are validated by FEM simulation, experiments, or both.

### 2.1. In-Plane Gravity Effects Mitigation

The main known cause for in-plane gravity effects on the stiffness of flexure pivots is their parasitic center shift. It is well-known that flexure pivots only approximate a rotational motion [10,15,16,22–25] and, in the case of a flexure-pivot oscillator, their center shift displaces the center of mass (COM) of the inertial body. The action of gravity on this displaced COM results in a torque that either contributes to the restoring torque of the flexures (positive stiffness contribution) or drives the oscillator away from its equilibrium (negative stiffness contribution). Therefore, in order to mitigate the effect of the center shift on the stiffness of the flexure-pivot time bases, two solutions are suggested: making the center shift small enough for the effect to be negligible (Section 2.1.1) or introducing a defect of opposite sign that counteracts this effect (Section 2.1.2).

### 2.1.1. Reducing the Parasitic Center Shift

This approach is already well-known, and the first flexure-pivot oscillator introduced in a watch [1,2], which is based on the cross-spring pivot [26,27], used the crossing ratio of the flexures that is known to minimize its center shift for this reason [22,23]. In previous work, we also presented the gravity-insensitive flexure pivot (GIFP), whose symmetrical arrangement of crossed flexures reduces the center shift for any crossing ratio [5]. Thanks to this, we showed that its rotational stiffness variation under in-plane load is of order 10 ppm, which corresponds to a chronometric precision of order 1 s/day. In contrast to previous publications, this paper evaluates the performance of this design approach directly in terms of chronometric accuracy, on mechanisms satisfying realistic watch specifications.

The co-RCC (Figure 1a) was designed such that each pair of *remote-center-of-compliance* (RCC) flexures, whose role is to guide the rotation, is connected in series to parallel flexures whose degree of freedom (DOF) is along the main component of the RCC flexures' center shift, that is, their bisector [6,7,23]. As a result, the center shift is drastically reduced in comparison to a standard RCC pivot. Experiments on a watch-scale (20 mm-diameter) silicon prototype with a frequency around 16 Hz (Figure 2) showed that this resulted in a chronometric stability within 10.4 s/day under varying gravity orientations (Figure 2b) [4]. The experimental method is fully described in [8]. The stability is expressed as daily rate (see [4] Equation (5.7)) from a reference frequency arbitrarily chosen as the mean frequency measured in horizontal position facing down (gravity in z+ direction). While this result is very close to the 10 s/day specification of the ISO 3159 norm for "Chronometer" watches, we will show that this can still be improved by using the design approach in Section 2.1.2 to reach the 5 s/day stability of "Master Chronometer" watches (Chronometer certificate: https://www.cosc.swiss/en/quality/precision accessed on 1 June 2022. Master Chronometer certificate: https://www.metas.ch/metas/en/home/dok/rechtliches/Zertifizierung_Uhren.html accessed on 1 June 2022).

An even greater center shift reduction is accomplished in the design of the TRIVOT oscillator (Figure 1b). Indeed, it is based on a theoretical kinematic design that theoretically has zero parasitic shift [10]. In practice, each *main flexure* is connected in series to parallel flexures having a radial DOF that absorbs the radial shortening of the main flexures as the pivot rotates. This has the consequence of reducing the center shift by one order of magnitude in comparison to the standard cross-spring pivot [10]. In order to provide realistic timekeeping accuracy data, a monolithic silicon TRIVOT oscillator satisfying mechanical watch specifications was designed. Figure 3 shows a part with a diameter of 15 mm, a height of 500 μm and a frequency of 25 Hz. The flexures have a thickness of 15 μm and all but the main flexures have their middle part rigidified to prevent buckling.

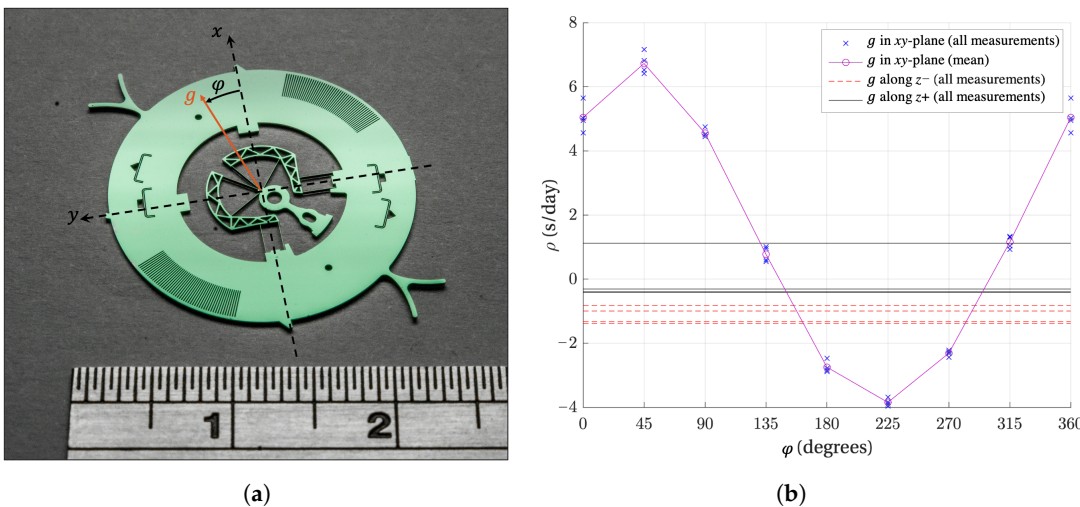

(**a**) (**b**)

**Figure 2.** Watch-scale silicon co-RCC oscillator (**a**) and its experimental daily rate $\rho$ (**b**) for varying angle $\varphi$ of the gravity load in the xy-plane (oscillation plane) and for gravity acting along the z axis (rotation axis). Four measurements were performed in each position [4].

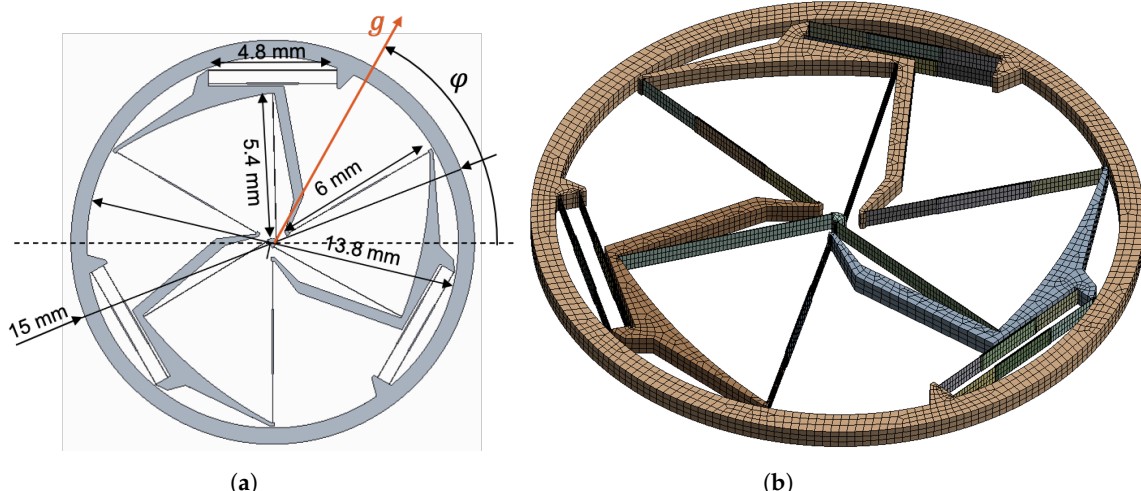

(**a**) (**b**)

**Figure 3.** Watch-scale silicon TRIVOT oscillator design: (**a**) main dimensions, (**b**) finite element model.

As this part could not be produced, FEM simulations were used to evaluate its frequency stability under varying gravity orientations (Figure 3b). The daily rate is derived from the frequency of the first vibration mode, where details of the FEM configuration can be found in [10]. We showed, in previous work, that this method gave reliable results and even overestimated slightly the defects (returned ±7.8 s/day error for the prototype in Figure 3 instead of the experimental ±5.2 s/day ([7] Section 7.7)). The results in Figure 4 show a frequency stability within 2.6 s/day for in-plane gravity effects, which is well within the specifications of the "Master Chronometer" certificate and shows the effectiveness of this design approach. Note, however, that the frequency for out-of-plane gravity loads (horizontal position) is far from the results for in-plane loads. We will show in Section 2.2 how this can be mitigated using the second step of our design methodology.

### 2.1.2. Offsetting the COM along the In-Plane Axis of Symmetry

For flexure-pivot architectures that have an axis of symmetry in the plane of motion, such as the co-RCC, the parasitic center shift is naturally symmetrical with respect to this axis. As a result, the stiffness (and frequency) variation in such oscillators for varying orientations of gravity in the oscillation plane is also symmetrical with respect to this axis. This can be seen from the daily rate variation measured on the co-RCC prototype for in-

plane gravity (Figure 2b), which exhibits a symmetry at 45 and 225 degrees corresponding to the axes of symmetry of the geometry (Figure 2a).

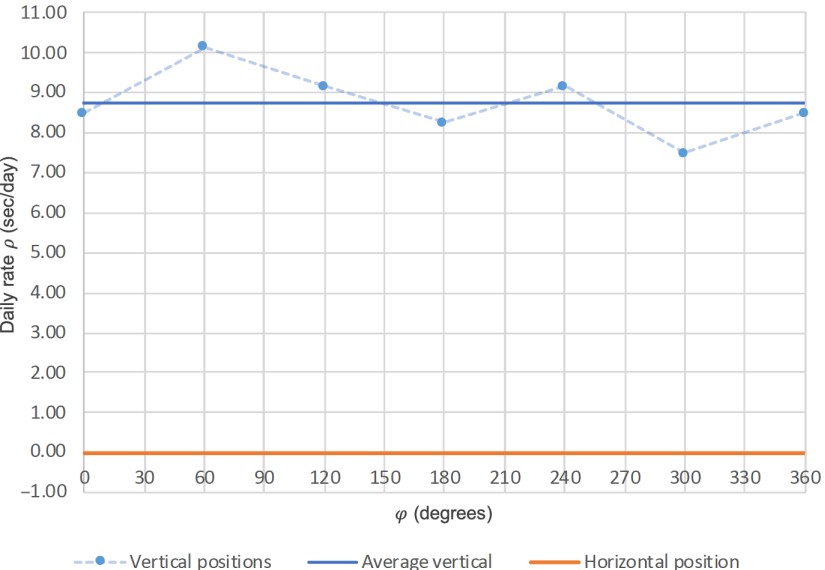

**Figure 4.** Daily rate $\rho$ of the watch-scale silicon TRIVOT oscillator for varying angle $\varphi$ of the gravity load in the oscillation plane (vertical positions) and for gravity acting along the rotation axis (horizontal position).

The sinusoidal shape of this defect is similar to the result of adding an ideal pendulum to a rotational spring, as depicted in Figure 5a. It is well-known that restoring the torque of the ideal pendulum with point mass $m$ and a massless rod of length $L$ is $mgL\sin\alpha$, where $\alpha$ is the angle with respect to gravity. When this pendulum is coupled to a rotational spring of stiffness $k_{el}$ having an equilibrium position at angle $\varphi$ with respect to gravity, the total restoring torque of the system for rotations $\theta$ from spring equilibrium is

$$M = k_{el}\theta + mgL\sin(\varphi + \theta) \tag{1}$$

which can be expressed using Taylor series expansion around $\theta = 0$ as

$$M = k_{el}\theta + mgL\left(\sin\varphi + \theta\cos\varphi + \mathcal{O}(\theta^2)\right) \tag{2}$$

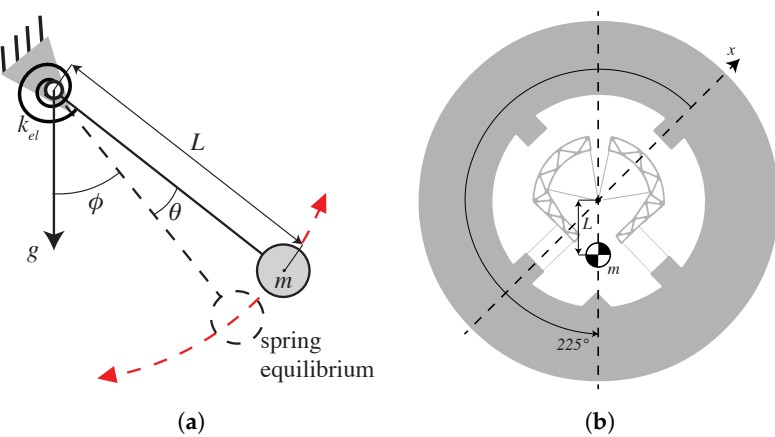

(**a**)          (**b**)

**Figure 5.** (**a**) Gravitational elasticity resulting from the addition of a pendulum to a rotational spring of stiffness $k_{el}$ and (**b**) practical implementation to compensate in-plane gravity effects on the co-RCC oscillator.

The resulting rotational stiffness for small displacements around the spring equilibrium (where higher order terms $\mathcal{O}(\theta^2)$ can be neglected) is

$$k_{tot} = dM/d\theta \approx k_{el} + mgL\cos\varphi \tag{3}$$

which corresponds to the shape of the defect observed in Figure 2b. Note that there is a 45 degree phase offset on the $\varphi$ parameter between the above formula and the experimental results. This means that the defect is similar to having a pendulum at 45 degrees from the part's reference frame, that is, along its axis of symmetry (Figure 2a).

Knowing how to create a defect similar to the one observed, we can create a defect of opposite sign to compensate it. This corresponds to adding a pendulum at 225 degrees, which, in practice, can be carried out by offsetting the COM of the part along its axis of symmetry by a distance $L$, as depicted in Figure 5b. This results in a negative gravitational elasticity when the gravity acts at an angle of $\phi = 45$ degrees and positive gravitational elasticity when at $\phi = 225$ degrees, which will compensate the change in frequency shown in Figure 2b. Using Equation (3) and ([4] Equation (5.7)), the resulting daily rate compensation can be estimated as

$$\Delta\rho = \mp 43200\frac{\Delta B}{k_{el}} \tag{4}$$

where $B = mgL$ is called the imbalance and $\Delta B$ is its variation used for frequency stability tuning ([4] Equation (6.8)).

In order to validate this design approach, prototypes with three different values of imbalance were fabricated, as can be seen from the difference in extra silicon mass on either side of their axis of symmetry (Figure 6). Note that the inner diameter of the inertial ring was also altered to keep the rotational inertia (i.e., the nominal frequency) constant. The imbalance value was calculated from their CAD model. Their oscillation frequencies were measured experimentally and simulated by FEM (as described previously) and Figure 7 displays their resulting frequency stability. Note that only the extreme positions (45 and 225 degrees) were simulated, as the other values will be in between. The experimental results were obtained by measuring two prototypes per type of variant (e.g., prototypes B13a and B13b).

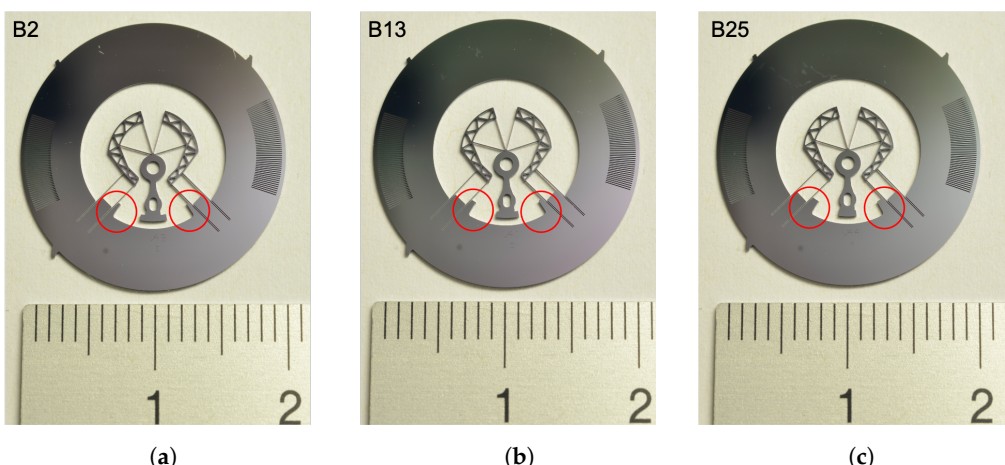

| (a) | (b) | (c) |

**Figure 6.** Silicon co-RCC prototypes with different values of imbalance caused by the mass variation in the symmetrical silicon outgrowths (circled on the figure): (**a**) prototype B2 with $B = 1.83$ nNm, (**b**) prototype B13 with $B = 12.6$ nNm, (**c**) prototype B25 with $B = 24.7$ nNm.

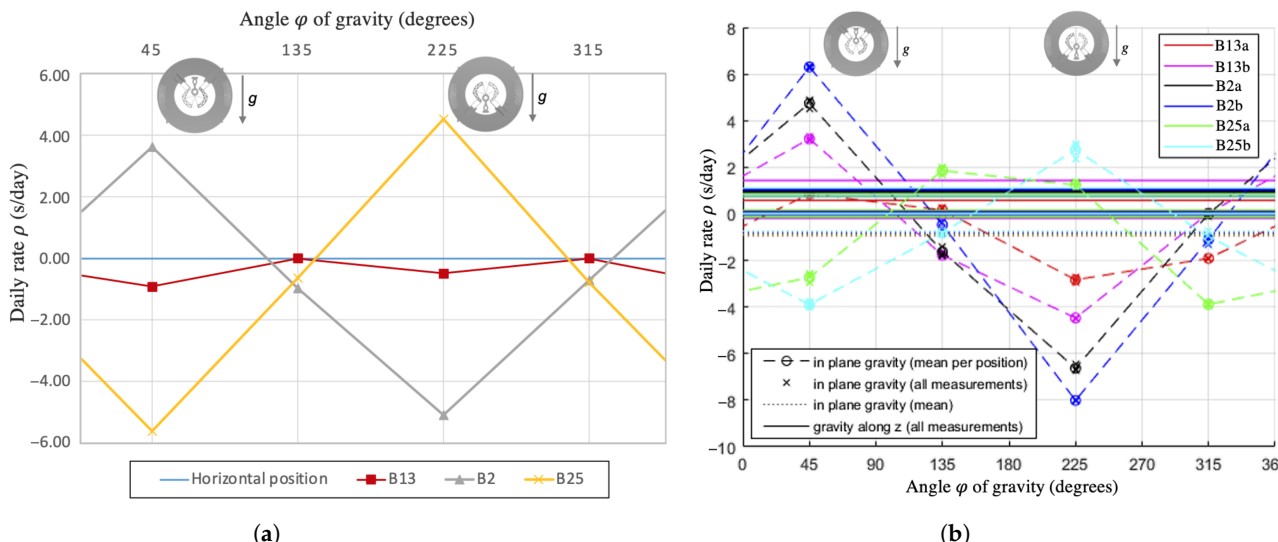

**Figure 7.** (**a**) FEM and (**b**) experimental validation of the effect of imbalance by varying the COM along the oscillator's axis of symmetry on its frequency stability for in-plane gravity orientations.

Table 1 summarizes the results and compares them to the compensating effect calculated with Equation (4). Note that the results of the tuning are not compared to the results of Figure 2, since elements other than the position of the COM changed between the prototypes. The main result here is the variation $\Delta\rho$ in in-plane daily rate obtained through the imbalance and not the absolute daily rate. The results show that offsetting the COM along the axis of symmetry of the oscillator enables to effectively impact and, hence, compensate in-plane gravity effects, without significantly influencing out-of-plane effects. Indeed, in all cases, the in-plane defect was decreased from prototypes B2 to B13 and could even be inverted from prototypes B13 to B25 (Figure 7b). This means that an imbalance value can be found for which the frequency stability can be virtually constant for all in-plane gravity orientations, as obtained with prototype B13 in FEM simulation (Figure 7a). Although the stability of our prototypes could still be improved by iterating on the imbalance value, prototype B13a reached an in-plane stability of 3.7 s/day, which is well within the "Master Chronometer" specification. It is worth noticing that this tuning method has practically no effect on the mean daily rate in vertical positions (in-plane gravity) nor on the daily rate for horizontal positions (out-of-plane gravity along *z*) (Figure 7b). This means that it can be used independently from our second design step (Section 2.2), which aims at reducing the difference between the mean daily rate in vertical and horizontal positions. The observed difference of approximately 1 s/day between both horizontal positions in the experimental results can be explained by the trapezoidal cross section of the flexures resulting from the DRIE manufacturing process ([4] Section 7.6.1). Note that this effect is not present in the FEM results, since rectangular cross sections were used in order to speed up computation. The thickness of this rectangular cross sections was, however, calculated to provide an equivalent bending stiffness to the trapezoidal cross section and oxide layer measured on the prototypes, see ([4] Equations (7.16)–(7.21)). Both the FEM results and the analytical calculations gave a good estimate of the compensation measured experimentally, hence providing a practical way of compensating such measured defects on flexure-pivot oscillators.

**Table 1.** Experimental, FEM and analytical daily rate compensation achieved by offsetting the COM of the co-RCC prototype along its axis of symmetry.

| Prototype | Imbalance $B$ | Analytical Daily Rate Compensation [1] $\Delta\rho$ | FEM Daily Rate $\rho$ | FEM Daily Rate Compensation [1] $\Delta\rho$ | Experimental Daily Rate $\rho$ | Experimental Daily Rate Compensation [1] $\Delta\rho$ |
|---|---|---|---|---|---|---|
| B2a | 1.83 nNm | - | ±4.4 s/day | - | ±5.69 s/day | - |
| B2b | 1.83 nNm | - | ±4.4 s/day | - | ±7.15 s/day | - |
| B13a | 12.6 nNm | ∓4.24 s/day | ∓0.46 s/day | ∓4.8 s/day | ±1.84 s/day | ∓4.56 s/day |
| B13b | 12.6 nNm | ∓4.24 s/day | ∓0.46 s/day | ∓4.8 s/day | ±3.85 s/day | ∓2.57 s/day |
| B25a | 24.7 nNm | ∓8.99 s/day | ∓5.1 s/day | ∓9.42 s/day | ∓2.87 s/day | ∓9.29 s/day |
| B25b | 24.7 nNm | ∓8.99 s/day | ∓5.1 s/day | ∓9.42 s/day | ∓3.32 s/day | ∓9.74 s/day |

[1] These values were calculated with respect to the mean daily rate of prototypes of type B2. For the analytical result, an elastic angular stiffness $k_{el} = 0.11$ mNm was used, based on a measured prototype frequency of 16.6 Hz and CAD-estimated inertia of $1.01 \times 10^{-8}$ kg·m$^2$.

### 2.2. Out-of-Plane Gravity Effects Mitigation

In the previous section, we showed two design methods that successfully improved the frequency stability of flexure-pivot oscillators for varying orientations of gravity in the oscillation plane, reaching the 5 s/day accuracy requirement of the "Master Chronometer" certificate. This is, however, not always the case for out-of-plane gravity orientations. For instance, in the case of the silicon TRIVOT oscillator design, simulations showed a difference of about 9 s/day between the mean in-plane gravity effect and gravity acting along the rotation axis (Figure 4). To address this issue, this section provides a design approach that allows to reduce this difference between in-plane and out-of-plane oscillation frequencies so that, when combined with the approaches of Section 2.1, a frequency stability satisfying the most demanding mechanical timekeeper requirements can be achieved for all orientations.

This approach is based on altering the loading conditions of the flexures providing the oscillator's spring stiffness in vertical positions, while scarcely impacting the loading conditions in horizontal positions. This way, the effective oscillator stiffness, which depends on the loading of the flexures (an effect known as stress stiffening), can be modified for in-plane gravity until it matches the stiffness in the horizontal position, resulting in a satisfactory frequency stability. In practice, this is achieved by offsetting the COM of the oscillator perpendicularly to its plane of motion. As intended, this generates transversal loads in the rotational flexures when gravity acts in the rotation plane, while having a negligible impact on the loads in the flexures when gravity acts along the axis of rotation. Displacing the COM perpendicularly to the plane of motion also has the advantage that it does not affect the moving inertia, thus not changing the oscillation frequency. Figure 8 illustrates this on the well-known parallel leaf spring stage. When the COM $G$ is in the plane of the mechanism (Figure 8a), the gravity force $P$ induces an out-of-plane bending moment in the flexures in horizontal positions, but only tension-compression loads in vertical positions. In contrast, when the COM $G$ is offset from the plane of the mechanism (Figure 8b), gravity induces an out-of-plane bending moment in the flexures in horizontal positions (as in Figure 8a), but also an out-of-plane bending moment in vertical positions. The COM offset distance $d$ from the plane of the mechanism can then be chosen such that the effect of the gravity load $P$ on the stiffness of the mechanism along its DOF ($k = F/x$ in the case of the parallel stage) is quasi-identical in vertical and horizontal positions.

In order to implement this approach on the TRIVOT oscillator of Figure 3, a variant was designed where the mobile and fixed parts are interchanged, allowing to attach a balance wheel at the center of the mechanism, with its massive part out of the plane of the flexures (Figure 9a). The balance wheel is made of a copper–beryllium alloy commonly used in watchmaking, has a mass of $m = 0.16$ g and an inertia of $J = 3.39 \times 10^{-9}$ kg·m$^2$.

The resulting oscillator has a frequency of 18.4 Hz and its frequency stability for horizontal and vertical positions calculated by FEM is depicted in Figure 9b. One can see that, in comparison to the frequency stability of the prototype with its COM in the flexure plane (Figure 4), the mean daily rate in vertical positions is now lower than in horizontal position, showing the effectiveness of this design approach. Offsetting the COM even more (Figure 9c) allows to decrease the vertical mean daily rate even more in comparison to the horizontal one. This confirms the method and one can presume that an offset can be found where the mean daily rates in vertical and horizontal positions are practically equal. In our case, the difference of 4 s/day in Figure 9b was deemed sufficient for our application. It is worth noticing that, although the COM has been offset, the thickness of the oscillator in this implementation is only 1.5 mm, which is compatible with standard mechanical-watch movement dimensions.

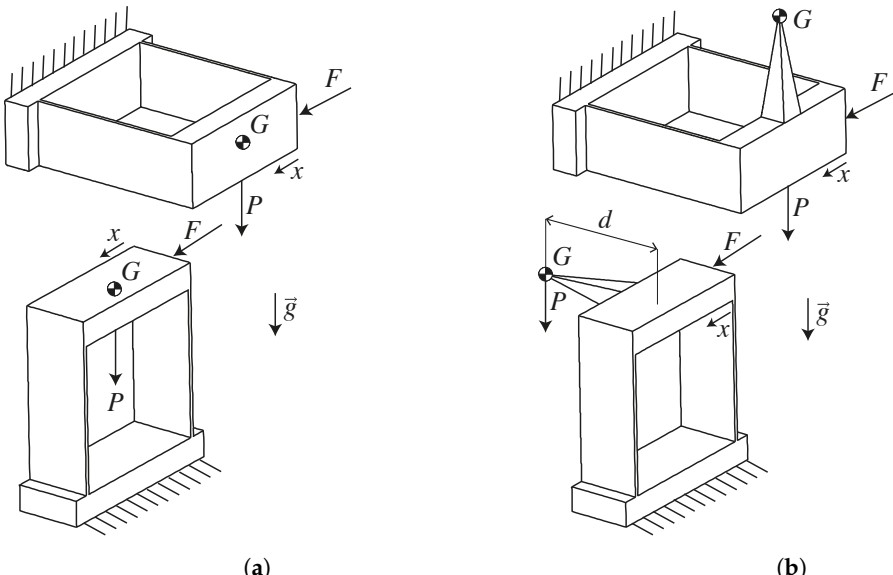

(**a**)                       (**b**)

**Figure 8.** Illustration of the effect of offsetting a flexure mechanism's COM from its plane of motion. When offsetting the COM (**b**), the gravity load in vertical position induces an out-of-plane bending moment $P \cdot d$ in the flexures that is not present when the COM is in the plane of motion (**a**). This, however, does not change the loading condition of the flexures in horizontal position so the parameter $d$ can be used to make the effects in vertical and horizontal positions match.

The fact that the rotational stiffness decreases in vertical positions when offsetting the COM from the rotation plane corresponds to the predictions of the analytical model for the cross-spring pivot under an out-of-plane bending moment ([28] Equation (6.28)). These leaf springs are subjected to the same boundary conditions as the main flexures of the TRIVOT, so we can expect the model to apply [10].

In order to validate this design approach experimentally, two weights were mounted to the silicon co-RCC oscillator using clamped dowel pins (Figure 10a). This way, when the weights are placed symmetrically with respect to the rotation plane, the COM of the mechanism is in this plane and, when they are both placed at the same distance on one side, different values of COM offset can be achieved by sliding the dowel pins (Figure 10b). With a combined mass of 18.5 mg per weight-pin assembly, a 157 mg silicon prototype and by offsetting the center of the 4 mm dowel pins by 0.85 mm and 1.85 from the mid-plane of the silicon part, *intermediate* and *large* COM offsets along $z$ of 0.28 mm and 0.47 mm, respectively, were implemented. The corresponding experimental daily rate measurements (Figure 10c) show that this design approach successfully manages to modify the difference between the mean daily rate in vertical and horizontal positions. Note that, for each offset value, the reference frequency (0 daily rate) is chosen as the mean frequency in horizontal position ($\vec{g}$ along $z$). It is also worth noticing that, while this is beyond the scope of this

study, offsetting the COM along z also reduced the frequency difference between the two horizontal positions, an unexpected but desirable consequence.

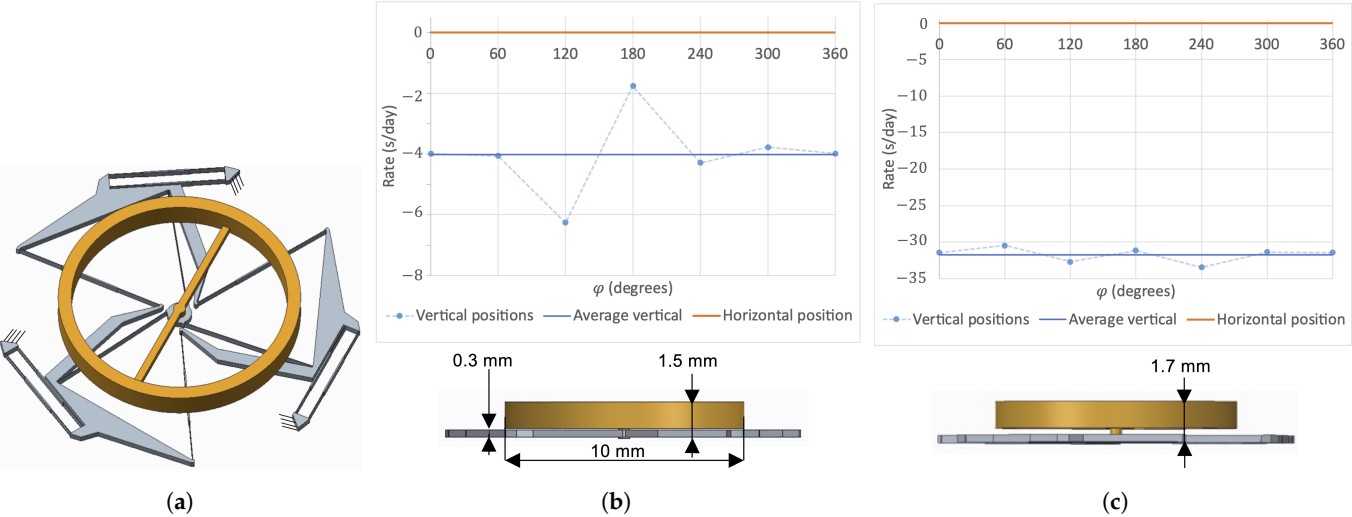

**Figure 9.** Watch-scale silicon TRIVOT oscillator (**a**) with COM offset from the flexure plane using a separate balance wheel and its daily rate for varying angle $\varphi$ of the gravity load in the oscillation plane (vertical positions) and for gravity acting along the rotation axis (horizontal position) with nominal COM offset (**b**) and increased offset (**c**).

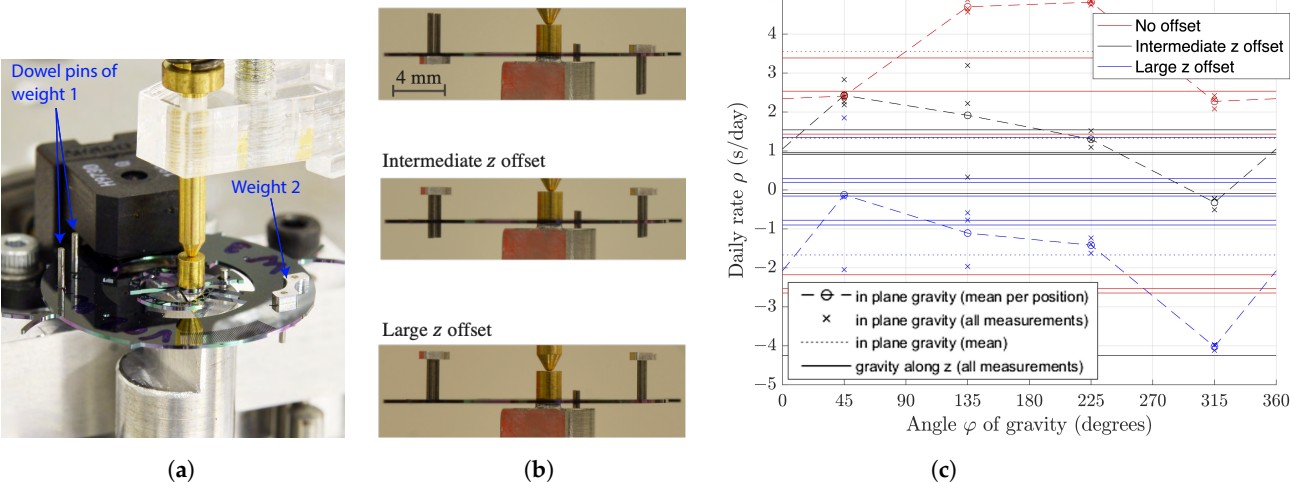

**Figure 10.** Experimental validation of the effect of offsetting the COM of a flexure pivot from its plane of motion using a co-RCC silicon prototype with external weights on slidable dowel pins: (**a**) prototype on test bench with symmetrical weights, (**b**) side view of the prototype with three different implementations of COM offset and (**c**) daily rate for varying orientations with respect to gravity for these three cases.

## 3. Conclusions

This article presented and validated two design approaches which, when combined, enable to compensate timekeeping accuracy defects of flexure-pivot oscillators in order to reach the strictest chronometric standards for mechanical watches. The effectiveness of these techniques was demonstrated on watch-scale silicon prototypes, which, due to the numerous other advantages in comparison to the traditional balance-hairspring oscillator [4–6], paves the way to their becoming the new standard in mechanical watchmaking. The two types of flexure-pivot oscillator architectures presented here correspond to the main categories observed in commercial and patented prototypes, namely, exhibiting either

an axis of symmetry like the co-RCC [2,29–32] or a ternary rotational symmetry like the TRIVOT [11,33–35]. Therefore, it can be assumed that the design approaches presented here apply to these architectures and provide a way of compensating gravity effects in numerous flexure-pivot oscillators. The gravity-compensation techniques discussed here are not limited to flexure time bases and can be applied to other compliant mechanisms, such as the parallel leaf spring translation stage in Figure 5b, where a constant stiffness is desired in varying orientations.

**Author Contributions:** Conceptualization, E.T. and S.H.; methodology, E.T. and S.H.; software, E.T.; validation, E.T. and Q.G.; formal analysis, E.T.; investigation, E.T.; resources, S.H.; data curation, E.T.; writing—original draft preparation, E.T.; writing—review and editing, E.T., Q.G. and S.H.; visualization, E.T. and S.H.; supervision, S.H.; project administration, S.H.; funding acquisition, S.H. All authors have read and agreed to the published version of the manuscript.

**Funding:** This research received no external funding.

**Institutional Review Board Statement:** Not applicable.

**Informed Consent Statement:** Not applicable.

**Data Availability Statement:** Not applicable.

**Acknowledgments:** The authors would like to thank Arnaud Maurel and Billy Nussbaumer for their assistance in the construction of the experimental setup.

**Conflicts of Interest:** The authors declare no conflict of interest.

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
