# Peer review of "Gravity-Compensation Design Approaches for Flexure-Pivot Time Bases"

_machines, doi:10.3390/machines10070580_

Round 1
Reviewer 1 Report
This paper introduces gravity compensation design approaches for flexure pivot time bases to increase chronometric stability. The concept is clever and has many applications not only in the watchmaking industry but also in other spring-based mechanisms. This paper is well written and arranged, however, I have a few minor questions/suggestions:
1) I suggest increasing the quality of Figure 8. I think that a hand-drawing image could reduce the quality of the paper.
2) The last term of equation (2) is not defined.
3) I suggest specifying the scale bar in Figure 10. And the weight 1 is hard to be recognized in Figure 10 (a).
4) The effect of offsetting the COM of a flexure pivot from its plane of motion is proven to increase the stability. However, the additional weights for offsetting could increase the volume of the mechanism. Could this design approach be applied to thin watch models?
Reviewer 2 Report
The paper presented two designs in which one was fabricated in real size, while the other was simulated. The presentation of the paper is a bit confusing and has to be restructured to make it flow easier.
Some comments are shown below:
1. Figure 8 has to be redrawn professionally.
2. Table 1 shows big deviation between the the FEA results on one hand and the analytical and experimental results on the other. Hence, the FEA model has to be improved to reduce the gap. Only then I can trust the FEA results which will be my only reference for the TRIVOT model that does not have an experimental component.
3. Line 238 instead of using "and axis", the author, I believe meant "an axis"
Round 2
Reviewer 2 Report
NA